# Dynamic Knowledge Integration in Multi-Agent Systems for Content Inference

**Atsushi Yamamoto**[*]
Honda Motor Co., Ltd.
Saitama, Japan
atsushi_01_yamamoto@jp.honda

**Takumi Iida**[*]
Elith Inc.
Tokyo, Japan
takumi.iida@elith.co.jp

**Taito Naruki**[*]
Elith Inc.
Tokyo, Japan
taito.naruki@elith.co.jp

**Akihiko Katagiri**
Honda Motor Co., Ltd.
Saitama, Japan
akihiko_katagiri@jp.honda

**Yudai Koike**
Honda Motor Co., Ltd.
Saitama, Japan
yudai_koike@jp.honda

**Ryuta Shimogauchi**
Elith Inc.
Tokyo, Japan
ryuta.shimogauchi@elith.co.jp

**Kota Shimomura**
Elith Inc.
Tokyo, Japan
kota.shimomura@elith.co.jp

**Eri Onami**
Elith Inc.
Tokyo, Japan
eri.onami@elith.co.jp

**Koki Inoue**
Elith Inc.
Tokyo, Japan
koki.inoue@elith.co.jp

**Osamu Ito**[†]
Honda Motor Co., Ltd.
Saitama, Japan
osamu_ito@jp.honda

## Abstract

Advancements in cutting-edge science and technology have resulted from the integration of multiple interdisciplinary domains beyond traditional academic boundaries. Achieving effective cross-domain knowledge-sharing and consensus-building is crucial. However, single-agent large language models (LLMs) solutions often struggle to integrate the diverse and highly specialized knowledge required in these contexts. This study proposes a multi-agent system with dynamic knowledge integration, where multiple specialized LLM-based agents cooperatively infer content by referencing different domain-specific databases. Each agent selectively and dynamically updates references based on conversational context to achieve deeper insight and more robust solutions. We propose four system architectures—*Decentralized*, *Centralized*, *Layered*, and *Shared Pool*—for agent coordination. We then evaluate these approaches on a title-to-abstract inference task using a subset of the arXiv dataset, demonstrating that multi-agent systems significantly outperform single-agent models in both accuracy and robustness. Notably, expert agents, restricted to domain-specific data, produce more precise and consistent outputs, and the *Decentralized* architecture fosters increased domain interaction. These findings suggest that the collaboration of specialized multi-agent systems can more effectively facilitate the consensus-building process in the advancement of complex interdisciplinary scientific domains.

---

[*]Equal contribution.
[†]Corresponding author.

# 1 INTRODUCTION

Rapid advancements in artificial intelligence (AI) technologies have been significantly transforming the methodologies of scientific discovery. One of the key drivers of this transformation is the evolution of large language models (LLMs) from single-agent generative models, which have demonstrated remarkable improvements in language capabilities, to multi-agent frameworks (Wu et al., 2023), where multiple agents collaborate to solve complex tasks. Multi-agent generative AI models have the potential to enhance problem-solving by deepening knowledge through structured discussions (Wang & Huang, 2024) and leveraging systematic decision-making processes (Song1 et al., 2024). Moreover, the integration of specialized knowledge using multi-agent systems has gained increasing attention. In particular, applications in the medical field have explored multi-agent systems that either rely on the inherent knowledge of LLMs (Tang et al., 2024a) or incorporate external knowledge expansions (Kim et al., 2024).

Addressing complex scientific challenges requires not only integrating interdisciplinary knowledge but also fostering structured interactions among domain experts. In real-world scenarios, research and development efforts involve experts from diverse fields, each possessing specialized knowledge, methodologies, and perspectives. However, cross-domain communication barriers and differences in organizational structures across departments often hinder effective collaboration. To address this, we propose a multi-agent system that reflects the communication structures across departments and enables dynamic knowledge integration. For effective discussions among domain-specific agents, structured datasets with appropriate labeling are crucial, as they enable agents to ground their reasoning in reliable, domain-specific knowledge. However, such datasets are often scarce or fragmented across different domains, limiting the potential for effective AI-driven collaboration. To address this challenge and evaluate the effectiveness of our approach, this study makes the following key contributions:

- We propose a multi-agent framework in which each agent references specialized domain databases and dynamically updates its contextual knowledge during conversation.

- We construct an evaluation dataset based on arXiv papers, which allows us to benchmark the system's cross-domain inference quality.

- We empirically compare different multi-agent integration strategies, examining both accuracy and stability.

Our approach is inspired by organizational structures, where different coordination patterns have evolved to optimize collaboration and decision-making. To this end, we design and analyze multiple agent coordination patterns—*Decentralized*, *Centralized*, *Layered*, and *Shared Pool*—each mirroring traditional hierarchies and communication patterns in human institutions (Figure 1). By leveraging these organizational structures, we aim to improve the adaptability and efficiency of multi-agent collaboration in research institutes and organizations.

# 2 RELATED WORK

## 2.1 LARGE LANGUAGE MODELS AND SINGLE-AGENT APPROACHES

Recent progress in large language models (LLMs) has led to substantial improvements across diverse natural language processing (NLP) tasks. GPT-3 (Brown et al., 2020), for instance, demonstrated the effectiveness of scaling model parameters to hundreds of billions, enabling strong few-shot performance without extensive task-specific supervision. Subsequently, efforts have been made to enhance the internal reasoning processes of LLMs. Chain-of-Thought (CoT) prompting (Wei et al., 2022) encourages the model to generate intermediate inferences, while ReAct (Yao et al., 2022) interleaves reasoning and actions, guiding the LLM to leverage external tools or sources during inference.

Despite these advances, single-agent LLMs still face significant challenges when multiple specialized domains must be integrated into a single solution. Large models often exhibit "hallucinations" or factual inconsistencies, particularly for domain-specific or cross-functional tasks demanding extensive specialized knowledge. In addition, single-agent paradigms are limited in their ability to incorporate multiple perspectives or facilitate a consensus among distinct expert viewpoints. Con-

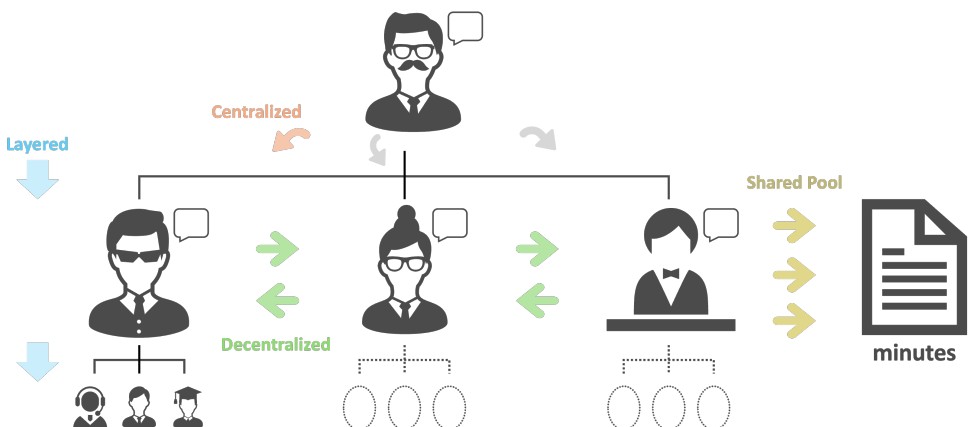

Figure 1: **A conceptual diagram illustrating our multi-agent system design inspired by organizational structures.** The figure demonstrates how different agent coordination patterns (*Decentralized*, *Centralized*, *Layered*, and *Shared Pool*) mirror traditional organizational hierarchies and communication patterns in human institutions. This biomimetic approach leverages established organizational principles to create more effective multi-agent collaboration.

sequently, there is increasing interest in multi-agent frameworks that can address these limitations by distributing roles and expertise among multiple, specialized agents.

## 2.2 COLLABORATIVE PROBLEM SOLVING VIA MULTI-AGENT SYSTEMS

A growing body of work focuses on orchestrating multiple LLM-based agents in pursuit of more reliable and comprehensive solutions. Multi-Agent Debate (Liang et al., 2024) is one such example, where agents engage in a structured discussion to critique each other's reasoning and reinforce logical consistency. This debate-like mechanism enables agents to detect and correct potential errors that might go unnoticed in single-agent setups. In a different vein, DyLAN (Liu et al., 2024b) introduces a dynamic agent network that allows for on-demand agent addition or removal, adapting the overall system architecture to the complexity of the target task.

In knowledge-intensive domains, MedAgents (Tang et al., 2024b) and MDAgents (Kim et al., 2024) both employ multiple agents with distinct areas of medical expertise. These studies have replicated complex medical decision-making and improved diagnostic accuracy by reflecting the collaborative methods used by experts in real-world clinical settings. Such findings highlight the potential for multi-agent approaches to facilitate specialized knowledge-sharing and collective decision-making, particularly when tasks necessitate in-depth expertise from diverse fields or departments.

## 2.3 DOMAIN KNOWLEDGE INTEGRATION AND RETRIEVAL-AUGMENTED GENERATION

RAG (Lewis et al., 2020) offers a way to mitigate the inherent limitations of LLMs by connecting them to external knowledge repositories. Instead of relying solely on parameters learned during pre-training, RAG pipelines query relevant documents in a vector database and incorporate the retrieved evidence into the generation process. REALM (Guu et al., 2020) further extends this concept by integrating retrieval and language modeling at the pre-training stage, improving factual accuracy and addressing the "knowledge cutoff" problem.

More recent methods explore how LLMs can dynamically invoke tools or APIs during inference. For instance, Toolformer (Schick et al., 2023) trains a language model to autonomously call specialized functions (e.g., calculators or translators) when needed. HuggingGPT (Shen et al., 2023) employs a collaborative system where a large language model (LLM) acts as a controller, orchestrating and integrating various expert models from the Hugging Face community to efficiently handle complex AI tasks across multiple domains and modalities. These approaches pave the way for more robust handling of domain-dependent information, aligning with the goals of multi-agent frameworks to leverage specialized knowledge.

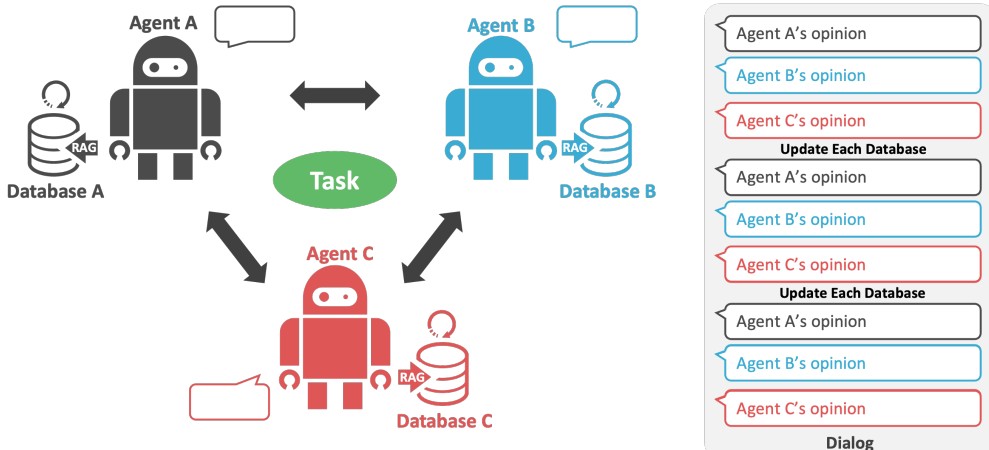

Figure 2: **Overview of the proposed approach.** Each agent is tailored to a specific domain or area of expertise and references a unique, specialized database of texts or other domain-relevant resources. As the conversation unfolds, all agents periodically update their references based on the latest contextual information. This ensures that the system remains responsive to newly surfaced insights or requirements, allowing for deeper domain-specific knowledge integration and a more robust final consensus.

## 2.4 MULTI-AGENT ARCHITECTURES

In parallel with algorithmic improvements, recent studies have examined architectural and design pattern considerations for multi-agent LLM systems. Magnetic-One (Fourney et al., 2024) similarly coordinates multiple sub-agents (e.g., web browsing, file manipulation, code generation) under a generalist framework, facilitating step-by-step task planning and execution.

Beyond centralized orchestration, researchers have begun to explore decentralized or hierarchical designs. For example, (Liu et al., 2024a) offers a survey of multi-agent design patterns, categorizing them into 18 architectural patterns. Furthermore, (Singh et al., 2025) also extracts common design patterns such as reflection, planning, and tool use. These multi-agent systems could better manage information flow. Motivated by these findings, we focus on four inter-agent structures—*Decentralized, Centralized, Layered, Shared Pool*—and incorporate dynamic knowledge updates tailored to domain expertise. This architecture aims to balance the benefits of specialized data references with flexible cross-agent collaboration, providing a foundation for more effective consensus-building among multiple expert agents.

## 3 METHOD

Our method addresses the challenge of integrating diverse domain knowledge through a multi-agent system. Each agent specializes in a particular domain, dynamically retrieving and updating relevant information as conversations progress. This approach enables both deep domain expertise and flexible cross-domain collaboration, which are essential for comprehensive problem-solving.

### 3.1 A MULTI-AGENT APPROACH WITH DYNAMIC KNOWLEDGE UPDATES

We propose a multi-agent architecture in which each agent references a unique, specialized database, updating its knowledge dynamically according to the state of the conversation (Figure 2). Whenever new information is introduced into the dialogue, each agent selectively queries its specialized data source for additional context. This design enables cross-domain discourse and prevents the system from being limited by a single static knowledge store. For example, if one agent references a specific concept, another agent can update its retrieval to cross-check or elaborate on that idea using its own domain expertise.

## 3.2 AGENT CONNECTION SCHEMES

We examine four inter-agent connection schemes (Figure 3): *Decentralized*, *Centralized*, *Layered*, and *Shared Pool*. Each agent has a distinct specialized domain database, but the communication flow differs:

**Decentralized.** Every agent communicates directly with all other agents. This dense interaction can trigger the emergence of novel cross-domain insights, though it may introduce more conversational overhead (e.g., a flat startup development team where all members engage in open discussions and participate in decision-making).

**Centralized.** A single "supervisor" agent directs the flow of conversation, querying each specialized agent as necessary. Through centralized management by a supervisor, this structure enables control of conversation flow and can lead to reasonable outcomes (e.g., a research laboratory where a single professor supervises the entire research process and students conduct specialized research).

**Layered.** The layered structure has a hierarchical form. Each agent independently thinks about and comments on problems, utilizing those insights in subsequent conversations. This enables the problem-solving process to be refined in stages, making it possible to reach higher-quality conclusions (e.g., a corporate R&D department where knowledge is systematically refined and integrated in stages).

**Shared Pool.** All conversation outputs are stored in a shared content pool, which every agent can access. This differs from the layered approach by allowing agents to consult all previous conversation content at each new step, maximizing information sharing at the risk of confusion (e.g., Wikipedia or a knowledge-sharing platform where previously recorded information is utilized to facilitate discussions).

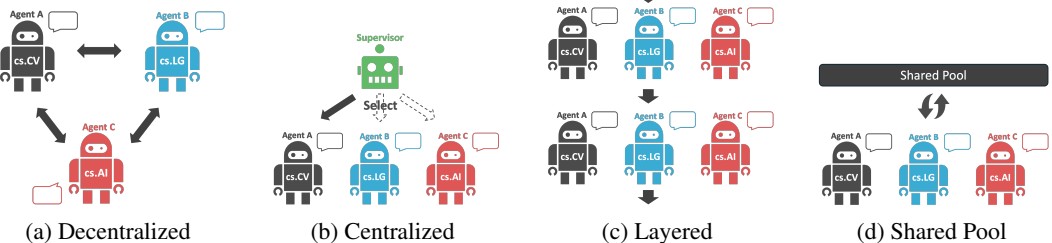

|  |  |  |  |
| --- | --- | --- | --- |
| (a) Decentralized | (b) Centralized | (c) Layered | (d) Shared Pool |

Figure 3: **Four different agent-connection structures examined in this study.** Each node represents an agent responsible for a particular domain or task, and directed lines indicate the flow of information during multi-turn dialogues. (a) *Decentralized* allows free bidirectional exchange among all agents, (b) *Centralized* routes communications through a single "supervisor" agent, (c) *Layered* uses a hierarchical structure, and (d) *Shared Pool* provides a communal repository of the entire conversation state. These configurations highlight trade-offs in communication efficiency, coordination complexity, and knowledge-sharing depth.

## 4 DATASET AND EVALUATION

### 4.1 ARXIV DATASET CREATION

To evaluate cross-domain inference, we select a paper-title-to-abstract prediction task based on an arXiv subset, which consists of preprints published on arXiv between January 1, 2023, and May 31, 2024. We use metadata such as *title*, *abstract*, *authors*, *categories*, and *sub-categories*. The data is partitioned into multiple domain-specific databases (e.g., `cs.CV`, `cs.LG`, `cs.AI`) so that each agent has access to a different subset. For testing, we choose papers that require references to at least two relevant domains.

## 4.2 EVALUATION PROTOCOL

Because full reconstruction of an abstract from only its title is intrinsically difficult (e.g., reconstructing numerical results or URLs is infeasible), we divide the abstract into two logical parts:

- *Background*
- *Approach*

Each part is evaluated separately using embedding-based cosine similarity between the generated text and the ground truth. This approach provides a more granular view of how well the system captures key elements of the source abstract.

## 5 EXPERIMENTS

### 5.1 EXPERIMENTAL SETUP

We compare multiple conditions (Figure 4):

- **Unified Single Agent (All-Domain).** A single agent has access to all domains (no specialization).
- **Single Expert Agent.** A single agent restricted to only the domains relevant to the target paper.
- **Distributed Multi-Agent (All-Domain).** Multiple agents each referencing a domain-specific database. Some agents may reference less relevant domains. We test the De-centralized, Centralized, Layered, and Shared Pool connection schemes.
- **Distributed Multi-Expert Agent.** Multiple specialized agents, but only those domains directly related to the current paper are included. Again, we test all four connection schemes.

For the multi-agent conditions, we define:

- *Round* = each attempt at generating the full abstract
- *Turn* = each exchange of conversation among the agents within a round

We fix the turn count to 4 and the round count to 20 for our main experiments. We use `gpt-4o-mini-2024-07-18` as the LLM for text generation and `stsb-xlm-r-multilingual` for vector embeddings in retrieval. In this setup, each abstract generation attempt involves 4 turns of inter-agent conversation, and this process is repeated 20 times.

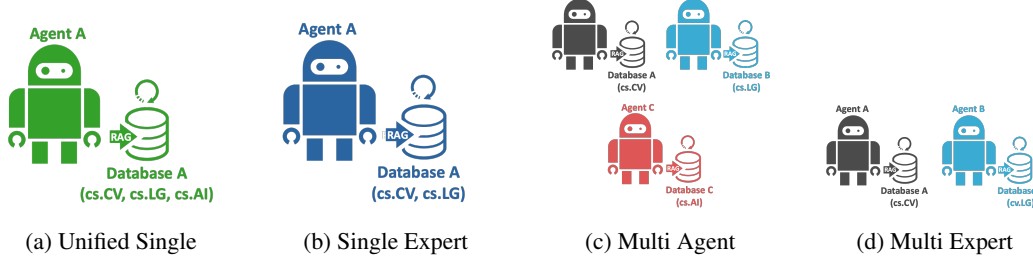

|     (a) Unified Single     |     (b) Single Expert     |     (c) Multi Agent     |     (d) Multi Expert     |

Figure 4: **Agent configurations.** (a) *Unified Single* employs one agent with unrestricted access to all domain-specific databases; (b) *Single Expert* narrows access to only the relevant domains for each test paper; (c) *Multi Agent* spawns multiple agents, each referencing one specialized database, though some may be less relevant; (d) *Multi Expert* similarly uses multiple agents but restricts them to databases deemed relevant for the target content.

Table 1: **Performance comparison of different agent configurations.** The table shows the mean cosine similarity (averaged over *Background* and *Approach* sections) and its standard deviation.

| Structure | Mode | Structure | Mean ↑ | Std ↓ |
|---|---|---|---|---|
| Single-Agent | All-Domain | – | 0.535 | 0.137 |
| | Expert | – | 0.552 | 0.129 |
| Multi-Agent | All-Domain | Decentralized | 0.566 | 0.106 |
| | | Centralized | 0.556 | 0.103 |
| | | Layered | 0.582 | 0.099 |
| | | Shared Pool | 0.565 | 0.096 |
| | Expert | Decentralized | **0.588** | 0.083 |
| | | Centralized | 0.571 | 0.097 |
| | | Layered | 0.579 | 0.106 |
| | | Shared Pool | 0.578 | **0.069** |

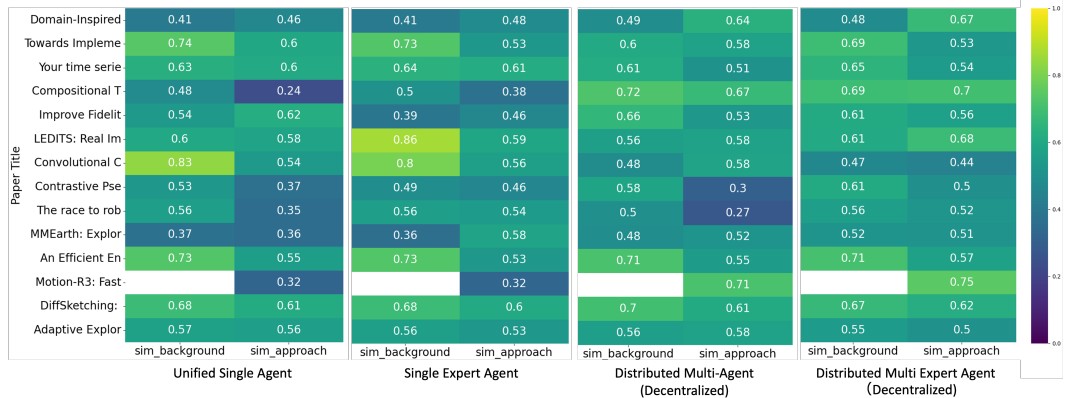

Figure 5: **Comparison of cosine similarity across all test papers for single-agent vs. multi-agent setups.** Each cell in the heatmap reflects the embedding-based cosine similarity between the *Background* or *Approach* portion of the ground truth and the system-generated text. Blank cells indicate abstracts that do not contain a distinct *Background* segment. Overall, multi-agent approaches (right side) yield higher similarities with lower variance, especially when domain specialization is applied (expert mode).

## 5.2 SINGLE-AGENT VS. MULTI-AGENT

Figure 5 shows a heatmap of cosine similarities between the background and approach sections of the ground truth abstracts and generated abstracts for each paper. These results indicate that the single-agent approach exhibits significant variability in output quality. Table 1 demonstrates that the standard deviation of the single-agent approach is higher than that of all multi-agent configurations. Conversely, the multi-agent systems achieve higher average cosine similarity scores and lower standard deviations compared to the single-agent approach, indicating more stable performance.

## 5.3 COMPARISON AMONG CONNECTION SCHEMES

In comparing connection structures in Table 1, *Layered* showed the highest average score in the fully distributed mode, while *Decentralized* achieved the best results in the distributed expert mode. All structures except the *Layered* demonstrated superior performance when using the expert mode, which only utilized databases from relevant fields. This suggests that processing is more effective when excluding agents and data from unrelated specialities. In the distributed expert configuration, structures with denser inter-agent connections tended to perform better. However, when non-specialist agents were included, the *Layered* with fewer connections showed the best results. *Shared Pool*, where all agents share the complete conversation history, achieved the most stable results with the lowest standard deviation under both conditions.

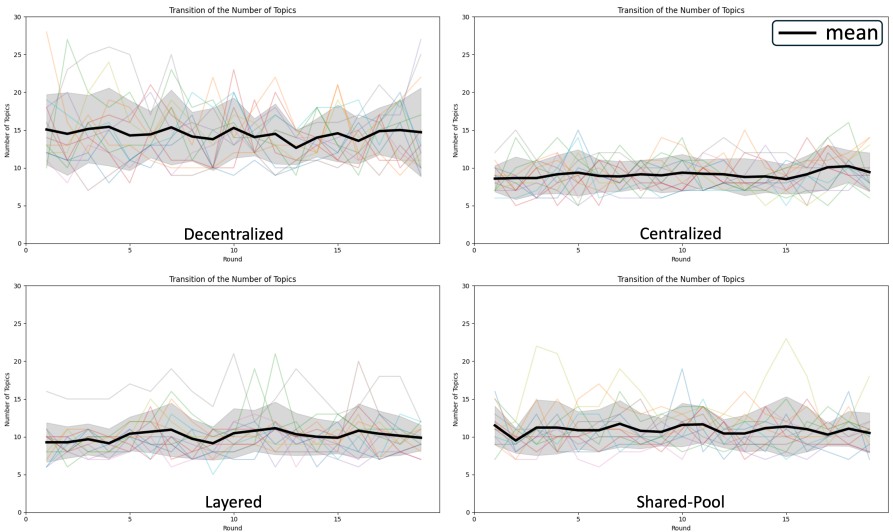

Figure 6: **Number of specialized (technical) terms appearing across rounds in Expert mode.** Each plot shows how many recognized domain-specific keywords or technical concepts were introduced per round of conversation. The gray shaded areas indicate confidence intervals based on standard deviation. While *Decentralized* fosters a high volume of specialized terminology exchange, *Centralized* keeps the conversation more streamlined, introducing fewer domain-specific terms.

## 5.4 ANALYSIS OF INTER-AGENT DIALOGUE

Figure 6 illustrates the number of specialized terms extracted (via `gpt-4o-mini-2024-07-18`) per round. The graph shows that the *Decentralized* contains significantly more technical terms compared to other methods. In contrast, *Centralized*, which selectively controls speakers, shows the lowest occurrence of technical terms.

These results suggest that while *Centralized* prioritizes conversation efficiency, it may limit the sharing of diverse perspectives and specialized knowledge. Conversely, *Decentralized*, which enables free dialogue between agents, promotes a broader exchange of specialized knowledge.

## 5.5 RELATIONSHIP OF COSINE SIMILARITY AND GENERATED RESULT

Table 2 presents two examples that highlight how the cosine similarity scores align with the quality of the generated abstracts. Although our system generates the entire abstract in one piece, we evaluate it by partitioning the text into two logical segments. In the first example, which achieved the best *Background* similarity under the *Single Expert* configuration, the generated text successfully reflects the core motivations and problem statements of the original. By contrast, the second example, corresponding to the worst *Approach* similarity in the *Unified Single Agent*, omits crucial methodological details and demonstrates a weaker connection to the ground-truth abstract. These observations suggest that higher cosine similarity generally corresponds to more faithful content reproduction, while lower similarity often indicates missing or ambiguous references to key elements.

## 6 ABLATION STUDY

### 6.1 EFFECTIVENESS OF DYNAMIC KNOWLEDGE UPDATES

We assess the contribution of dynamic knowledge updates to the overall performance of our multi-agent system. Each agent is capable of updating its retrieved information based on the evolving conversation context. This dynamic updating mechanism ensures that agents continually refine their internal knowledge base, thereby incorporating the latest and most relevant domain-specific insights into the dialogue.

Table 2: **Qualitative examples comparing ground truth vs. generated abstracts.** "BG Sim" and "App. Sim" refer to the cosine similarity scores for the *Background* and *Approach* parts, respectively.

| Title | GT Abstract (Excerpt) | Generated Abstract (Excerpt) | BG Sim | App. Sim |
|---|---|---|---|---|
| *LEDITS: Real Image Editing with DDPM Inversion and Semantic Guidance* | [Background] Recent large-scale text-guided diffusion models provide powerful image-generation capabilities. Currently, a significant effort is given to enable the modification of these images using text only as means to offer intuitive and versatile editing. However, editing proves to be difficult for these generative models due to the inherent nature of editing techniques, which involves preserving certain content from the original image. Conversely, in text-based models, even minor modifications to the text prompt frequently result in an entirely distinct result, making attaining one-shot generation that accurately corresponds to the users intent exceedingly challenging. In addition, to edit a real image using these state-of-the-art tools, one must first invert the image into the pre-trained models domain - adding another factor affecting the edit quality, as well as latency. [Approach] we propose LEDITS - a combined lightweight approach for real-image editing, incorporating the Edit Friendly DDPM inversion technique with Semantic Guidance, thus extending Semantic Guidance to real image editing, while harnessing the editing capabilities of DDPM inversion as well. | [Background] Recent advancements in large-scale language-image models, particularly text-guided diffusion models, have significantly enhanced the ability to generate photorealistic images, leading to a surge in intuitive image editing techniques. However, existing methods often struggle with user-friendliness, generalization across diverse domains, and maintaining high fidelity to the original images. [Approach] This paper aims to address these challenges by introducing LEDITS, a novel method for real image editing that leverages Denoising Diffusion Probabilistic Model (DDPM) inversion and semantic guidance. The authors claim that their approach provides a superior balance between editability and fidelity, allowing for precise modifications such as changing the color of specific objects while preserving their original characteristics. | 0.86 | 0.59 |
| *Compositional Text-to-Image Generation with Dense Blob Representations* | [Background] Existing text-to-image models struggle to follow complex text prompts, raising the need for extra grounding inputs for better controllability. [Approach] Based on blob representations, we develop a blob-grounded text-to-image diffusion model, termed BlobGEN, for compositional generation. Particularly, we introduce a new masked cross-attention module to disentangle the fusion between blob representations and visual features. | [Background] Recent advancements in visual synthesis have raised concerns about the potential for generating harmful images, such as those promoting hate or violating privacy. This paper aims to address the gap in research regarding the transformation of harmful images into responsible ones through a novel task called responsible visual editing. The authors propose a Cognitive Editor (CoEditor) that utilizes a large multimodal model to identify and modify abstract concepts within images while minimizing alterations. [Approach] The approach involves a two-stage cognitive process: first, a perceptual cognitive process to determine what needs modification, and second, a behavioral cognitive process to strategize the modification. | 0.48 | 0.24 |

To evaluate this mechanism, we conducted an ablation study comparing the performance with and without dynamic knowledge updates across various agent connection schemes (Table 3). The results indicate that enabling dynamic updates generally leads to improvements in the mean cosine similarity.

For instance, in the *Decentralized* configuration, enabling dynamic updates improved the mean cosine similarity from 0.578 to 0.588, and it's the best performance. The most pronounced effect was observed in the *Layered* configuration, where the mean performance increased from 0.542 to 0.579 when dynamic updates were applied.

Collectively, these results validate the effectiveness of dynamic knowledge updates in enhancing both the accuracy of our multi-agent system.

## 6.2 NUMBER OF ROUNDS AND TURNS

Figure 7 shows the results of varying rounds and turns for the expert system. The results demonstrate remarkable robustness across different round and turn counts, consistently outperforming single-agent baselines. This robustness indicates that the multi-agent system's advantages stem from its architecture rather than specific parameter settings. However, we observe slight performance degradation when the number of rounds or turns becomes excessive, suggesting an optimal operating range. This indicates that increasing dialogue iterations does not necessarily improve performance—instead, excessive interaction can cause discussions to lose focus and potentially diminish output quality.

Table 3: **Performance comparison with and without dynamic knowledge updates in Expert mode.** Each row shows the mean cosine similarity (Mean) and standard deviation (Std) for a specific connection scheme when updates are disabled or enabled. Dynamic updates generally improve overall performance, highlighting their importance in multi-turn conversations.

| Structure | Dynamic | Mean↑ | Std↓ |
|---|---|---|---|
| Decentralized |  | 0.578 | 0.082 |
|  | ✓ | 0.588 | 0.083 |
| Centralized |  | 0.567 | 0.096 |
|  | ✓ | 0.571 | 0.097 |
| Layered |  | 0.542 | 0.098 |
|  | ✓ | 0.579 | 0.106 |
| Shared Pool |  | 0.578 | 0.094 |
|  | ✓ | 0.578 | 0.069 |

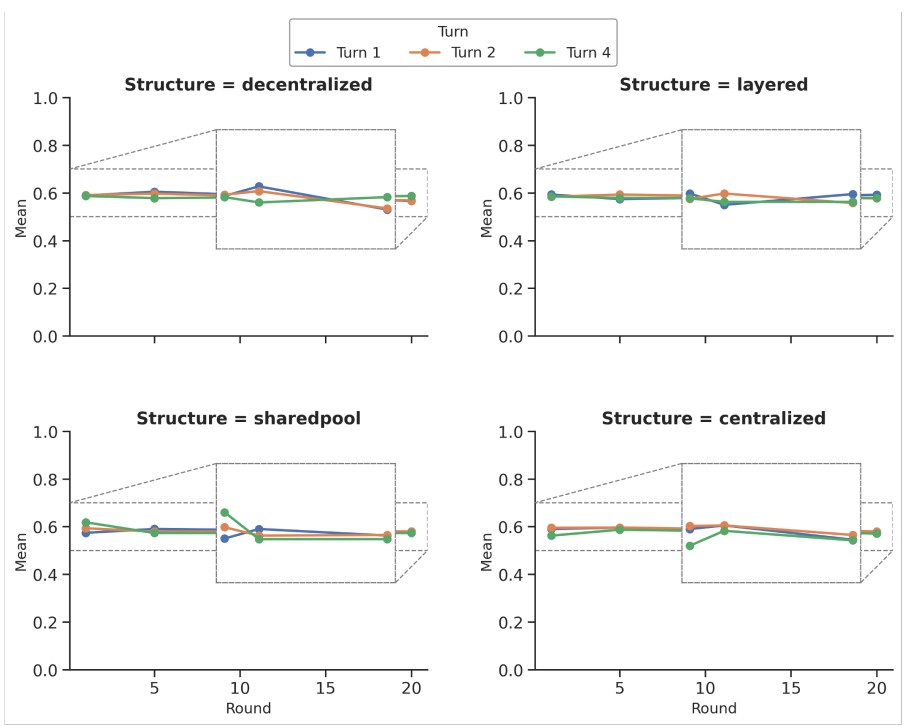

Figure 7: **Progression of cosine similarity with varying numbers of rounds and turns in Expert mode.** The vertical axis shows the mean cosine similarity score, while the horizontal axis indicates the number of rounds.

## 7 CONCLUSION

We have presented a multi-agent system that dynamically integrates knowledge across multiple domain-specific databases. Experimental results on an arXiv-based title-to-abstract task show that this approach outperforms single-agent baselines in both accuracy and stability. Restricting agents to relevant domain data (Expert mode) further improves performance, indicating the importance of specialization. Among the four connection schemes, *Decentralized* encourages more diverse inter-agent communication, while *Layered* can be advantageous when some agents reference irrelevant domains. *Shared Pool* provides stable performance with minimal variance.

Our findings suggest that multi-agent collaboration is a promising approach for integrating specialized knowledge and facilitating consensus-building in scientific discussions within interdisciplinary research domains. Future work includes evaluating scalability to more extensive domain sets and exploring adaptive mechanisms that can automatically switch between connection schemes based on task complexity.

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

# A  APPENDIX

## A.1  ANALYSIS OF TECHNICAL TERM DISTRIBUTION

To further analyze the patterns of technical term usage across different connection schemes, we visualized the distribution of specialized terms using heatmaps per paper. The heatmaps show the occurrence of technical terms across conversation rounds for each connection scheme.

The heatmaps in Figures 9 and 8 illustrate this distribution for two example papers across different connection schemes. The vertical axis represents conversation rounds, while the horizontal axis shows individual technical terms. The intensity of each cell indicates how frequently a term was used in that particular round.

## A.2  UMAP VISUALIZATIONS OF REFERENCED PAPERS

In this section, we present UMAP visualizations that show which papers were referenced by each multi-agent (`cs.AI`, `cs.LG`, `cs.CV`), alongside the ground-truth abstract ("True Abstract"). These visualizations help us understand how each agent's domain-specific database was utilized throughout the conversation rounds.

Figures 10 and 11 highlight two example papers using the four connection schemes (Decentralized, Centralized, Layered, and Shared Pool). Each circle represents a referenced document, where the color indicates the domain, the circle size corresponds to the number of rounds in which the document was referenced, and the darkness of the color shade indicates the frequency of references. The red star denotes the ground-truth abstract itself. By examining these plots, we can observe how strongly each agent relies on its domain-specific resources and how frequently such resources are reused across multiple rounds of conversation.

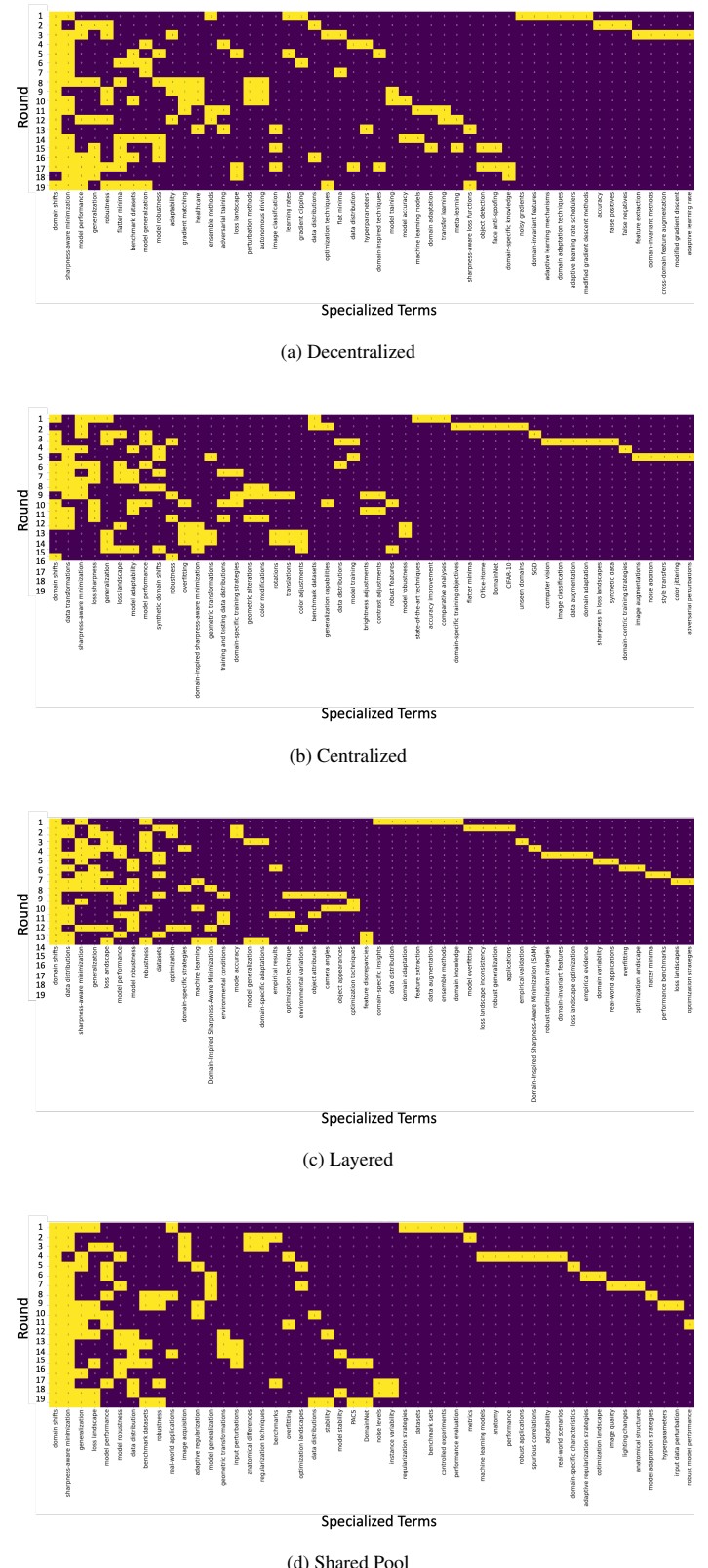

(a) Decentralized

(b) Centralized

(c) Layered

(d) Shared Pool

Figure 8: **Per-round distribution of specialized terms for the paper "Domain-Inspired Sharpness-Aware Minimization Under Domain Shifts" in Expert mode.** The vertical axis indicates the round number, while the horizontal axis represents specialized terms appearing at each round. Each subfigure corresponds to a different multi-agent connection scheme.

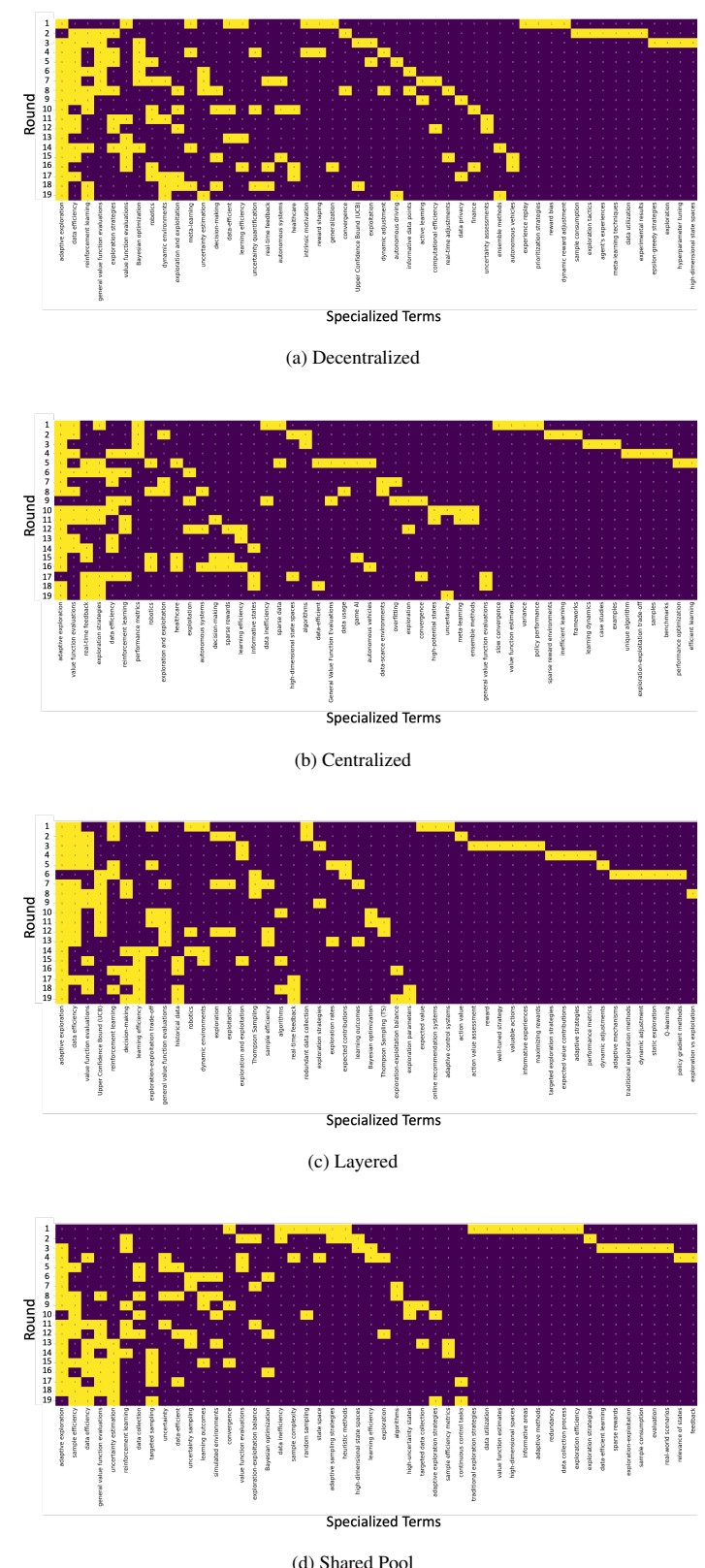

(a) Decentralized

(b) Centralized

(c) Layered

(d) Shared Pool

Figure 9: **Per-round distribution of specialized terms for the paper "Adaptive Exploration for Data-Efficient General Value Function Evaluations" in Expert mode.** The vertical axis indicates the round number, while the horizontal axis represents specialized terms appearing at each round. Each subfigure corresponds to a different multi-agent connection scheme.

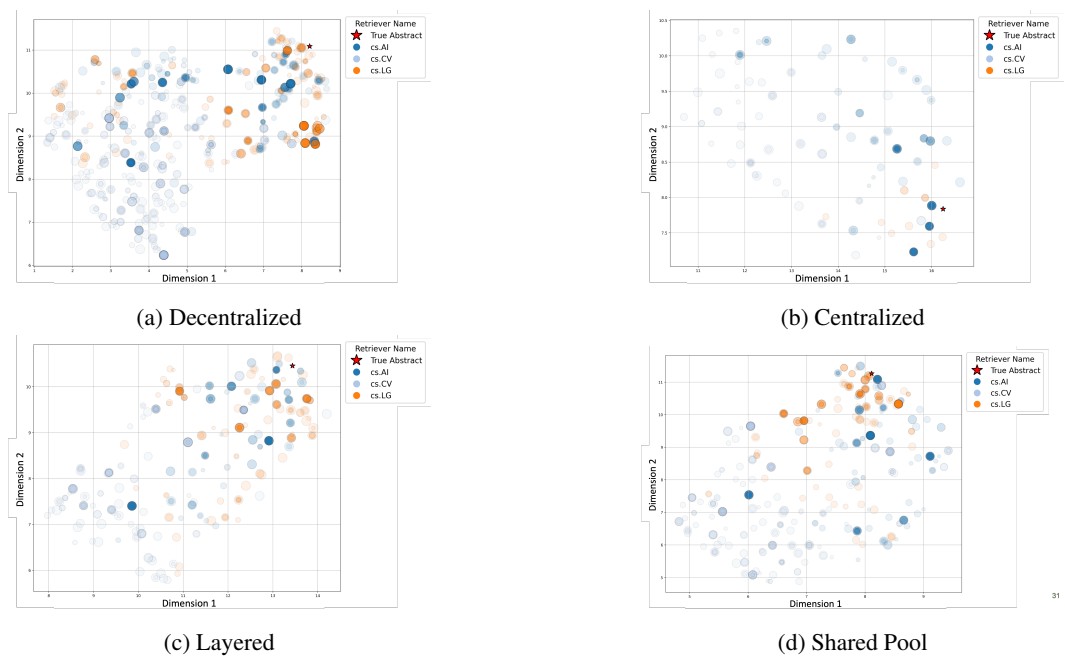

Figure 10: **UMAP Visualization for "Adaptive Exploration for Data-Efficient General Value Function Evaluations."** Circles represent referenced documents, colored by domain (cs.AI, cs.LG, cs.CV). The star marker indicates the true abstract. Circle size reflects the number of rounds in which each document was referenced, and darker hues indicate more frequent references.

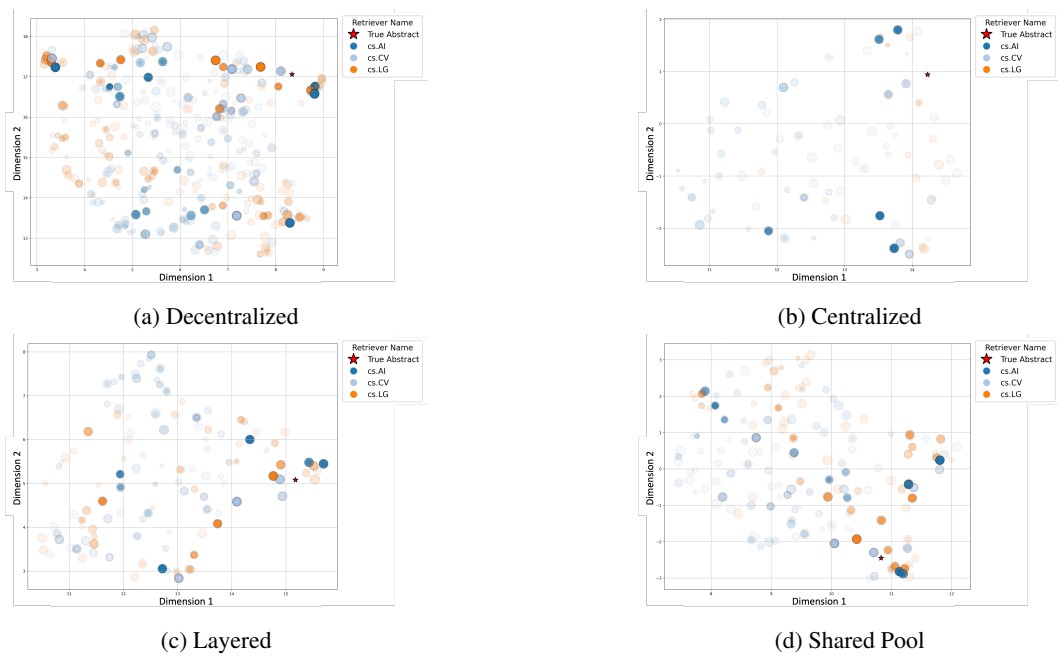

Figure 11: **UMAP Visualization for "Domain-Inspired Sharpness-Aware Minimization Under Domain Shifts."** Circles represent referenced documents, colored by domain (cs.AI, cs.LG, cs.CV). The star marker indicates the true abstract. Circle size reflects the number of rounds in which each document was referenced, and darker hues indicate more frequent references.

