# OpenReview forum: "Dynamic Knowledge Integration in Multi-Agent Systems for Content Inference"
_ICLR.cc/2025/Workshop/AgenticAI — ICLR 2025 Workshop AgenticAI Poster_

### Official Review · Reviewer_yg2E · 2025-02-28
**Good work**

**Rating:** 8
**Confidence:** 4

**Review:**

Summary: This paper presents a multi-agent system that dynamically integrates knowledge across domain-specific LLM agents for scientific content inference. The study examines four agent coordination architectures (Decentralized, Centralized, Layered, and Shared Pool) and evaluates them on arXiv title-to-abstract generation.

Strengths:
- Well-motivated approach that reflects real-world organizational structures
- Strong empirical performance that multi-agent systems outperform single-agent models
- Insightful comparisons across different coordination architectures
- Useful ablation studies demonstrating the importance of dynamic knowledge updates

Weaknesses:
- Limited theoretical analysis explaining why certain architectures perform better
- Dataset construction details could be more thorough
- Comparisons with state-of-the-art RAG-based models would strengthen evaluation
- Insufficient discussion of practical applications beyond academic paper generation

---

### Official Review · Reviewer_vChF · 2025-02-28
**Good work**

**Rating:** 6
**Confidence:** 4

**Review:**

This paper works on integrating diverse and highly specialized knowledge through multi-agent systems. Four system architectures are presented for agent coordination. Detailed experiments and comprehensive analysis are conducted to compare between these four structures. In general, the paper is well written and the methodologies are innovative.

quality: The paper is in good quality with a clear presentation to show the background and challenges in multi-agent coordination across different knowledge domains. The methodologies behind this paper are comprehensive followed by complete experiments and insightful analysis. The paper is in a good flow and the writing is formal and clear.

clarity: The paper clearly shows the proposed system structures that are important to agent coordination. The comprehensive and rigorous experiments show the effectiveness of the proposed coordination structures. Moreover, the analysis is also thorough and insightful.

originality: The methodologies proposed in this paper are novel and important for applying multi-agents in AI systems.

Pros:
1. The problem of multi-agent coordination is important and the methodologies shown in this paper are novel.
2. The experiments are comprehensive and rigorously prove the effectiveness of the proposed architectures. The analysis is also complete and insightful.
3. The paper is well-written and in a clear flow.

Cons:
1. The mechanism behind the coordination is not clearly presented and analyzed.
2. The datasets are limited in domains, and the knowledge gap between different agents is not presented which makes it unclear whether the improvement comes from the coordination of knowledge or an ensemble from different agents.

---

### Official Review · Reviewer_1sQM · 2025-03-03
**Clear presentation but the scope is somewhat narrow**

**Rating:** 5
**Confidence:** 5

**Review:**

Summary

This paper introduces a multi-agent system that dynamically integrates domain-specific knowledge for improved cross-domain content inference. The authors evaluate four agent-coordination architectures—Decentralized, Centralized, Layered, and Shared Pool—using an arXiv-based title-to-abstract inference task. Experimental results suggest that multi-agent systems consistently outperform single-agent models in both accuracy and robustness, and the authors conclude that specialized multi-agent collaboration can more effectively facilitate consensus-building in interdisciplinary settings.




Strengths

1. Well-Structured： The paper is well organized, with a clear presentation and smooth language flow, making it easy to read.

2. Detailed Experiments and Analysis： Under both single-agent and multi-agent configurations, the authors conduct experiments on four different agent coordination strategies and provide comparative analyses. They also perform ablation studies on dynamic knowledge updates, as well as on the effect of varying the number of rounds and turns, thereby offering deeper insights into the effectiveness of each approach.


Weaknesses

1. Task Specificity
All experiments and evaluations focus solely on a title-to-abstract task, leaving it uncertain how well the findings generalize to other text-generation or content-inference scenarios.

2. Lack of SOTA Comparison
It is not entirely surprising that multi-agent systems outperform single-agent baselines, given that multiple agents have access to more data and consume more computational power. A comparison with other state-of-the-art (SOTA) methods would better situate the paper’s contributions within the broader research community.

3. Limited Contribution
Although the paper compares four coordination strategies, these approaches do not introduce any major innovation for multi-agent collaboration. Moreover, the experiments rely on a single dataset and a single task, limiting the generalizability of the findings. The four strategies also exhibit inconsistent performance in both the All-domain and Expert structures, making it difficult to provide definitive guidance for other studies or real-world applications. As a result, the paper’s overall contribution appears limited.

4. Lack of Computational Cost Analysis
Including a comparison of runtime and resource usage under different coordination strategies would strengthen the paper’s persuasiveness and practical relevance.

5. Simplified Evaluation Metric
The authors rely heavily on cosine similarity between generated segments and the ground-truth text. While this offers a coarse measure of alignment, it may not capture the more nuanced aspects.

The paper presents a clear exploration of multi-agent coordination for title-to-abstract generation, but its scope is somewhat narrow, and it would benefit from stronger comparisons, broader tasks, and additional analyses (e.g., computational costs and more sophisticated evaluation metrics).

---

### Decision · Program_Chairs · 2025-03-05

Accept (Poster)